# Adolescent and Juvenile Idiopathic Scoliosis: Which Patients Obtain Good Results with 12 Hours of Cheneau–Toulouse–Munster Nighttime Bracing?

**DOI:** 10.3390/children9060909

**Published:** 2022-06-17

**Authors:** Gautier De Chelle, Virginie Rampal, Imad Bentellis, Arnaud Fernandez, Carlo Bertoncelli, Jean-Luc Clément, Federico Solla

**Affiliations:** 1Paediatric Orthopaedic Unit, Lenval Foundation, 57, Avenue de la Californie, 06200 Nice, France; dechelle.g@pediatrie-chulenval-nice.fr (G.D.C.); rocher-rampal.v@pediatrie-chulenval-nice.fr (V.R.); carlo.bertoncelli@univ-cotedazur.fr (C.B.); clement.jluc@wanadoo.fr (J.-L.C.); 2Urology Unit, CHU, 30 Voie Romaine, 06000 Nice, France; bentellis.i@chu-nice.fr; 3Children’s Psychiatry, Lenval Foundation, 57, Avenue de la Californie, 06200 Nice, France; fernandez.a@pediatrie-chulenval-nice.fr

**Keywords:** idiopathic scoliosis, nighttime bracing, brace, CTM, Cheneau brace, predictive factor, indication

## Abstract

Background: The results of 12 h nighttime Cheneau–Toulouse–Munster (CTM) brace wear on adolescent idiopathic scoliosis are poorly described. Objective: The main objective was to analyze the efficiency of 12 h nighttime CTM brace wear on adolescent idiopathic scoliosis. The secondary objective was to identify the factors influencing good results. Methods: One hundred and fifty consecutive patients treated between 2006 and 2017 were retrospectively analyzed with subgroup analysis for the main curve pattern (main thoracic or main lumbar). The inclusion criteria were evolutive scoliosis, 12 h nighttime CTM brace wear, Risser stages 0-1-2 at the time of the prescription, and Cobb angle below 45 degrees. Success was defined as no surgery, and the main curve Cobb angle (CA) progression ≤5°. The overcurve was defined as the proximal thoracic curve above the main thoracic and mid-thoracic above the main lumbar curves. A logistic regression model was built to assess the predictors of success. RESULTS: Overall success was 70%: 60% for main thoracic (MT) and 84% for main lumbar scoliosis (ML) (*p* = 0.003). Efficacy was 62% at Risser stage 0 and 78% at Risser stage 1–2 (*p* = 0.054). For MT, failure was associated with high in-brace sagittal C7 tilt (Odds Ratio = 0.72, *p* = 0.014) and low initial overcurve CA (Odds Ratio = 0.42, *p* = 0.044). For ML, a high standing height was associated with success (OR = 1.42, *p* = 0.035), and frontal unbalanced C7 tilt was associated with failure (OR = 0.43, *p* = 0.02). Conclusion: Twelve-hour nighttime CTM brace wear provided good results for main lumbar curves with balanced frontal C7 tilt. For MT, this treatment is indicated if the in-brace sagittal C7 tilt is well balanced from Risser stage 2.

## 1. Introduction

The main goal of bracing adolescent and juvenile idiopathic scoliosis (AJIS) during pubertal growth is to arrest the progression of the Cobb angle (CAP) [1,2,3,4]. There is no worldwide consensus on the type of brace and wear time. The Cheneau–Toulouse–Munster (CTM) brace is widely used in Europe with full-time wear. The effectiveness of full-time brace wear is approximately 80% for all types of idiopathic scoliosis [1,2,3,4].

Observational studies have shown that scoliosis behavior (worsening or stability) depends on the curve pattern, with thoracic curves being more evolutive than lumbar curves [5,6]. Sanders and Al proposed a modified Lenke classification (mLenke) separating non-surgical scoliosis in main thoracic curves (MT) and main lumbar curves (ML) [6], the latter including both thoracolumbar and lumbar main curves.

According to Weinstein’s BrAIST study, the most common duration of brace wear is >18 h daily. However, the authors reported a 90% success rate for a minimum of 12.9 h of effective brace wear. [7]

Often, the actual and prescribed times of brace wear per day may differ [8]. Compliance is higher with part-time brace prescriptions: 65% compliance for 23 h prescriptions, 71% for 12 h prescriptions, and 94% for 8 h prescriptions [9]. Katz et al. found similar success ratios of brace wear time between 23 h and 16 h in-brace prescriptions but found poorer results for a wearing time down to 12 h (71% of success) and <7 h (31%) with a Boston brace [10]. However, the literature remains contradictory about this [11,12]. To the best of our knowledge, no studies have been published featuring a prescription of 12 h nighttime wear of the CTM brace.

We hypothesized that 12 h of nighttime CTM brace wear is effective in treating AJIS. The main objective was to describe brace efficacy based on different curve patterns and the Risser classification. Afterward, we aimed to identify clinical and radiological factors associated with good results.

## 2. Materials and Methods

### 2.1. Patients

This was a monocentric retrospective cohort study of consecutive patients treated in a pediatric orthopedic surgery unit between 2006 and 2017 by a single pediatric spine surgeon (JLC).

Inclusion criteria were: evolutive AJIS (CAP > 5° between 2 successive roentgenograms or Cobb angle >30° from the first visit), Risser sign = 0, 1 or 2, and initial prescription of part-time CTM brace. Incomplete medical files without available full spine roentgenograms, as well as clinical and radiological control 1 year after weaning, were excluded.

### 2.2. Treatment

The CTM Brace is a tailored, underarm, single piece, balanced, asymmetric brace with a pressure pad and expansion chambers composed of 4 mm polyethylene (Figure 1).

Data were collected at 4 dates:-Date 1: Medical prescription, 3D scan body acquisition followed by brace-digitalized fabrication. Initial brace wear occurs at day 30 with a target of 12 h wear time out of 24, meaning all night and a small part of the day, usually the evening;-Date 2: In-brace roentgenograms at Day 60, after foam pad adjunction to enhance correction;-Date 3: Weaning, having a Risser test ≥ 4 and height stability;-Date 4: Medical visit to evaluate brace efficiency with medical examination and roentgenograms, at least one year after weaning.

For brace renewal, delivery and foaming were simultaneous. Monitoring was shared as follows: orthotist visits every 2–3 months and medical visits every 6 months. No physiotherapy was prescribed. 

### 2.3. Measures and X-ray Analysis

One experienced pediatric spine surgeon (JLC) performed clinical measurements in successive medical visits. These measurements included standing and sitting height, rib hump and lumbar prominence measured with lead wire, and inequality of leg length search for dates 1, 3 and 4. Non-compliance, such as failing to respect the 12 h wear time or not wearing the brace at all, was assessed through interrogation.

Full Spine Roentgenograms were taken on dates 1, 2 and 4 with EOS^®^ technology on nude spine after a night without the brace, to avoid maintaining proprioceptive effect. In-brace full spine roentgenograms were taken on date 2. 

An independent physician (GdC) performed digital measurements on “Keops Analyser” (Smaio, Lyon, France). This software has proved its superiority over direct measurements on X-rays [13]. Data were classified in 3 categories: GLOBAL for global spine parameters, REGIONAL for scoliotic deformation pattern, and LOCAL for apex vertebrae.

Lumbar lordosis (LL) and thoracic kyphosis (TK) were measured using the Cobb method and represent angles of vertebral segments in lordosis or kyphosis, respectively, independently from predefined levels [14]. The overcurve was defined as the proximal to the thoracic curve above the main thoracic and mid-thoracic above the main lumbar curves. Frontal C7 tilt was the angle between the vertical line passing through the center of the sacrum and the line passing through the center of C7 and the center of the sacrum. Sagittal C7 tilt angle was measured between the vertical line passing through the center of the sacrum and the line passing through the center of C7 and the center of the sacrum.

The acromio-femoral angle was the angle between the bi-coracoid line and the bi-femoral line, passing through the center of both femoral heads. The lateral displacement of the apical vertebra was measured as the distance from the apical vertebra and the midline passing through the center of the sacrum. Ilio-lumbar angle was measured between the bi-iliac line and the tangent of the most inclined lumbar vertebra. Wedging was measured as the angle between the superior and inferior vertebral plates. (Appendix A)

Full data are available in Appendix A.

### 2.4. Main Outcome

The primary endpoint for brace success was defined if two criteria were present: (1) no indication for surgery and (2) the main curve CAP ≤ 5° between date 1 and date 4.

### 2.5. Statistical Analysis

Data were managed by an independent physician (GDC), and statistical analysis was carried out by an independent statistician (IB).

Chi squared and exact Fisher’s test were used to compare qualitative values, and a T-test was used for quantitative values after verification of normal distribution. Boxplots were made to describe repetitive measurements. To identify the cutoff values, ROC curves were realized. A logistic regression model was used for the primary outcome. Subgroup analysis was performed using the mLenke classification and Risser test.

The multiple regression model included potential influencing factors with clinical pertinence and *p* < 0.2 from univariate analysis. Odds ratios (OR) are presented with confidence interval 95% (CI) and *p*-values.

R software was used to perform calculations and diagnostic analysis was performed on Anger University software.

### 2.6. Ethical Statement

All procedures described in this article were in accordance with the ethical standards of the institutional review board (CNIL with number 2017728v0, 14 September 2017) and with the 1964 Helsinki Declaration and its later amendments. Formal consent from parents and patients was not necessary for the retrospective analysis of the anonymized data.

## 3. Results

### 3.1. Population

Of all 742 patients screened, 150 were included (Figure 2).

The median initial Cobb angle was 26°, with 74% of patients between 15 and 29° (Table 1).

The mean age was 12 years and 9 months, with 88% girls. Fifty-one percent of adolescents were classified as Risser stage 0, 28% as Risser stage 1 and 21% as Risser stage 2.

### 3.2. Main Outcome and Coronal Angles

The overall success rates was 70%, with 60% and 84% (*p* = 0.003) for MT and ML, respectively (Table 2).

Concerning Risser grades, success was 62% for patients in stage 0, and 78% for patients in stages 1 and 2 (*p* = 0.054). Overall, 21 patients needed surgery, and 66% of them were in the MT group. Risser stages were similarly distributed in the MT and ML groups (Table 3):

About the main curve Cobb angle, a decrease of its mean was observed after weaning in the whole cohort and in subgroups. Furthermore, the variance was higher for dates 3 and 4 compared to dates 1 and 2 (Figure 3).

### 3.3. Sagittal Angles

A lower TK was found for MT curves, and its relative mean loss was 15% during brace treatment (Figure 3). LL remained stable.

### 3.4. Predictive Factors

#### 3.4.1. Date 1: Orthosis Prescription Day

For clinical data (Table 4), higher standing and sitting height were associated with success for ML. The lower thoracic rib hump was weakly associated with success for MT (15 mm vs 12.5 mm, *p* = 0.09), as was lumbar prominence with ML. No association with gender or unequal leg length was found.

Regarding X-ray data,

Global Spine Measures: The increase in C7 tilt was correlated with failure in the frontal plane for ML (3.6° vs 1.3° *p* = 0.002). An increase in the acromio-femoral angle was correlated with failure for the MT (2.7° vs 1.6° *p* = 0.02).

Regional Spine Measures: Low main-CA < 25° was associated with success for MT. Furthermore, subgroup analysis for CA severity showed an inversion of the success rate between 15–29° and 30–44° for MT, 66% and 43%, respectively, whereas it was 86% and 73% for ML. ROC curves were plotted to determine the initial CA cutoff, while 24° was the cutoff for MT (sensibility 0.82, specificity 0.48). No cutoff was found for ML. Lower overcurve-CA was significantly associated with success for ML (12.6° vs 4.1° *p* = 0.021) and MT (17.4° vs 3.4 *p* = 0.034).

Local Spine Measures: Higher frontal apex vertebrae lateral displacement was associated with failure for the main lumbar curve (33 mm vs 22 mm *p* = 0.019). Higher apex vertebrae wedging was found in the failure group.

Pelvis Measures: Higher pelvic tilt was correlated with success for MT (62 vs 9.5, *p* = 0.045).

#### 3.4.2. Date 2: In-Brace X-ray 

Global Spine Measures (Table 5): An imbalance of sagittal C7 tilt for MT was linked with failure (4° vs 2° *p* = 0.001).

Regional Spine Measures: Low main-CA was associated with success for MT and ML. Its in-brace reduction was correlated with success in both MT and ML, with the reduction in success groups being 46% and 66%, respectively.

Local Spine Measure: Differences in ML for apex lateral displacement were not statistically significant.

Pelvis Measures: No links were found.

#### 3.4.3. Compliance

Twelve percent of our patients were non-compliant, and 55% of them were in the failure group (*p* = 0.025).

#### 3.4.4. Multivariate Analysis 

For MT, the most important factor associated with failure was unbalanced in-brace sagittal C7 tilt with OR = 0.72 IC95%: (0.54; 0.91) *p* = 0.0139. Higher overcurve-CA was strongly associated with failure, with OR = 0.42 IC95%: (0.54; 0.91), *p* = 0.0437 (Table 6).

For ML, unbalanced in-brace frontal C7 tilt was associated with failure (OR 0.43, IC95%: 0.19, 0.83, *p* = 0.0196), and higher standing height was associated with success (OR 1.42 IC95%: 1.07, 2.12, *p* = 0.0348). (Table 7).

## 4. Discussion

Our study confirmed the efficiency of 12 h of nighttime brace wear, with higher success for ML curves [14,15].

The effectiveness of full-time CTM brace wear was reported in three different studies with 81 to 83% success [2,3,4]. From our data, the overall success rate was 70%, but only 21 out of 150 patients (14%) underwent surgery or would need surgery in the future, with CA ≥45°. In turn, 86% of our cohort avoided surgery completely.

In the BrAIST study, the mean initial CA was 30°, and success was defined by CA ≤ 50° at skeletal maturity. A 90% success rate has been reported with a duration of daily brace wear between 12.9 and 17.6 h [7]. With the same criteria of CA ≤ 50°, Thompson and Al found differences of success between MT (66%) and ML (85%) using TLSO full-time brace wear; these ratios of success are consistent with our results of 60% and 84% success with part-time bracing, while considering that we have used more severe criteria of success [14].

Eight hours of hypercorrective brace wear (e.g., CAEN Brace) during the night has been prescribed for single curves with 66 to 75% success [16,17,18,19]. These results are comparable to our series of CTM braces, where we included both single and double curves. We emphasized nighttime because growth is higher during the night and because wearing a brace during the night is easier for a teenager than at school or during the daytime, in general.

We found better results for adolescent idiopathic scoliosis (70% success) than for juvenile idiopathic scoliosis compared to the Jarvis study (43% success for nighttime bracing). [20] Boulot and Lateur recommend early treatment of AIS with nighttime bracing. [3,21] However, from our study, the earlier we treat with part-time bracing, the worse results we have, and our results are inferior to full-time brace wear for Risser stage 0–1 in MT and Risser stage 0 in ML. This result deserves caution since we tend to treat the more severe curves earlier.

Concerning the stage of growth, our data did not suggest that the Risser sign is a strong predictor of success. Thus, a more precise growth classification than the Risser sign should help define the protocol of brace wear, as well as find the curve acceleration phase. Correlations have been demonstrated using hand radiographs with the Tanner–Whitehouse-III or Greulich and Pyle atlas. Sander’s classification should also be useful [6]. For clinical data, peak height velocity remains the best indicator, whereas the Tanner stage is an “invasive” examination for teenage girls in routine consultation.

For MT scoliosis, a Cobb angle of 28 degrees was the maximum limit when wearing a brace for 12 h overnight, which was consistent with Lateur’s study [21] (Figure 4).

Among less common predictors, the sitting height, thoracic rib hump, presence of high overcurve CA and main curve poor reduction in-brace were associated with failure for both MT and ML.

For MT, a good sagittal C7 tilt balance in-brace was associated with success. However, various patterns of the main thoracic curves are classified in the same gross group, with different responses to brace treatment [22]. This finding suggests the brace prescriptor to use lateral bending with assessment of the sagittal alignment to obtain better CA reduction in brace [23].

For ML, despite our small sample size, unbalanced frontal C7 tilt was correlated with failure. This imbalance is always on the same side as the lateral displacement of the apex vertebrae. This encourages us to use the lumbar shift with posterolateral force as much as possible for correcting lumbar and thoraco-lumbar curves.

For both groups, overcurve management is an obstacle to success, even with our underarm brace. 

This study has several limitations. The sample size was limited; we were not able to include a considerable number of patients who had incomplete roentgenograms in their medical files. Moreover, we were not able to measure vertebral rotation. Furthermore, brace compliance was established only through patient and parent questioning and environment data, and no in-brace monitor was used.

Despite these limitations, our population is comparable to other studies. All clinical and radiological data has been analyzed with rigorous methodology: strict selection of cases, constant treatment protocol, x-rays under brace to check the effectiveness of the brace, and radiological results one year after brace weaning.

## 5. Conclusions

For main lumbar curves, treatment was effective for patients presenting with well-balanced scoliosis with frontal tilt C7 < 1 cm. Therefore, we suggest treating such curves with 12 h of CTM brace wear nightly.

For main thoracic curves, the success rate was lower. Good results are expected from Risser stage 2 if the sagittal tilt of C7 and the overcurve are low.

## Figures and Tables

**Figure 1 children-09-00909-f001:**
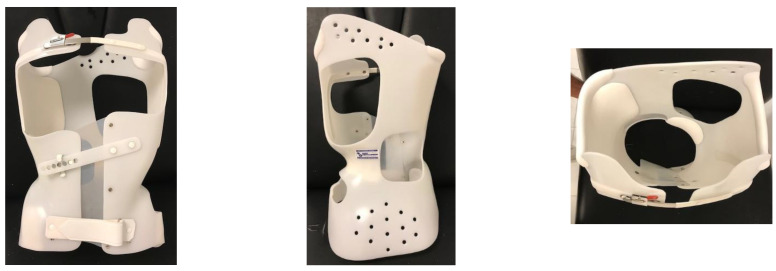
Cheneau–Toulouse–Munster Brace.

**Figure 2 children-09-00909-f002:**
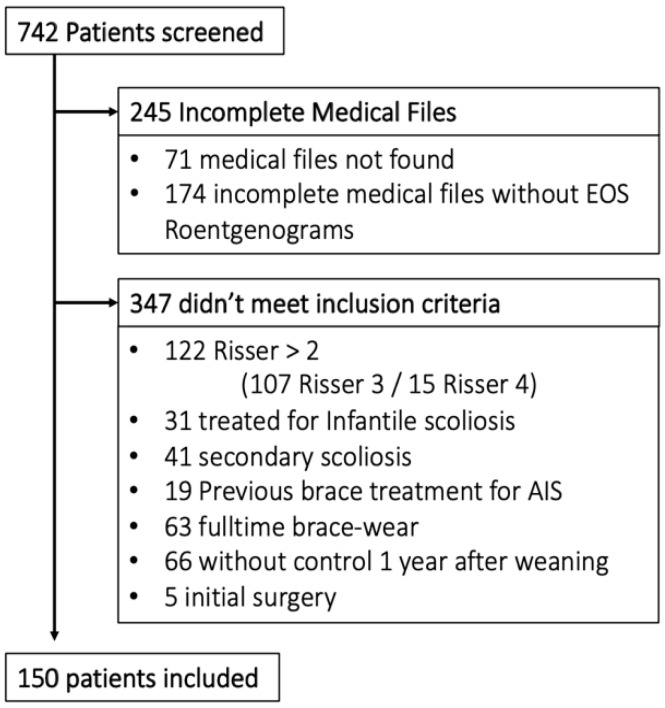
Flowchart.

**Figure 3 children-09-00909-f003:**
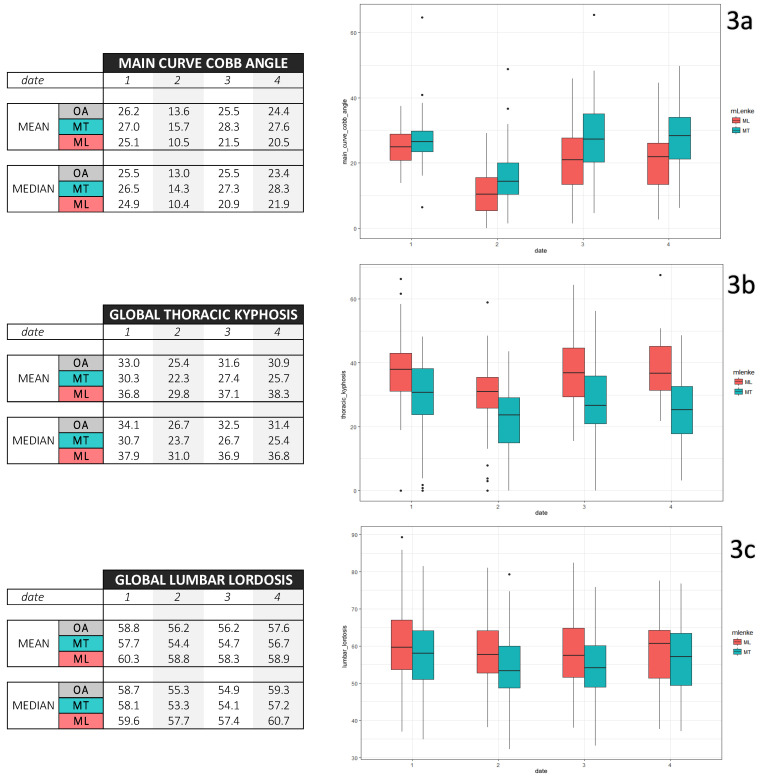
Descriptive repetitive EOS roentgenogram measures (mean, median and boxplots) for date 1 brace prescription day, date 2 in-brace roentgenogram, date 3 weaning and date 4 control 1 year after weaning. (**a**) concerns main curve Cobb angle, (**b**) concerns thoracic kyphosis, and (**c**) concerns lumbar lordosis.

**Figure 4 children-09-00909-f004:**
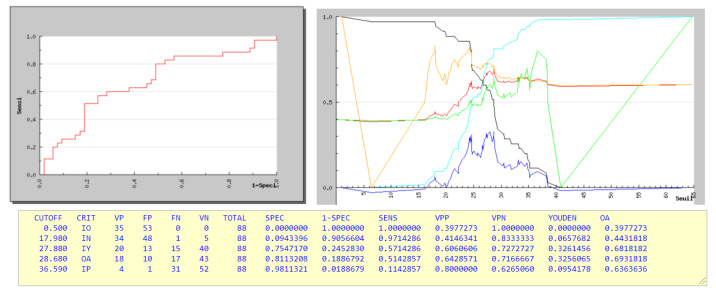
ROC Curves and Threshold Curves for initial date 1 Cobb Angle for Main Thoracic scoliosis. Black: sensibility; Blue: Youden Index; Red: OA; Green: Positive Predictive Value; Orange: Negative Predictive Value; Cyan: Specificity.

**Table 1 children-09-00909-t001:** Overall population.

			150
CLINIC	Age year (median [IQR])		13 (12, 14)
	Sex (%)		130 F (87.7%) 20 M (13.3%)
	Standing Height cm (median [IQR])		156 (151, 161)
	Sitting Height cm (median [IQR])		80.00 (77, 84)
	Thoracic Gibbosity mm (median [IQR])		10.00 (10, 20)
	Lumbar Gibbosity mm (median [IQR])		5 (5, 10)
	Unequal Leg Lengh (%)		21 (14)
X-RAY	Risser Test (%)	0	77 (51.3)
		1	42 (28)
		2	31 (20.7)
	mLenke (%)	ML	62 (41.3)
		MT	88 (58.7)
	Main Curve Cobb Angle degree (median [IQR])		25 (22, 29)
		<15°	2 (1.3)
	Main Curve Cobb Angle	15–29°	111 (74)
	n (%)	30–44°	36 (24)
		>45°	1 (0.7)
	Pelvic Incidence degree (median [IQR])		47.6 (6, 38)
	Sacral Slope degree (median [IQR])		39 (34, 46)
	Pelvic Tilt degree (median [IQR])		8 (3, 13)
	Lumbar Lordosis degree (median [IQR])		59 (52, 64)
	Thoracic Kyphosis degree (median [IQR])		34 (26, 41)

**Table 2 children-09-00909-t002:** Efficiency of brace treatment—overall and subgroups. **Bold indicates *p* < 0.05**.

		Failure	Success	% Success	*p*
Overall		45	105	70	
mLenke	MT	35	53	60	**0.003**
	ML	10	52	84	
Risser Test	0	29	48	62	0.109
	1	9	33	79	
	2	7	24	77	
Risser Test	0	29	48	62	0.054
	1 + 2	16	57	78	

**Table 3 children-09-00909-t003:** Cross analysis of subgroup efficiency.

		MT	ML
		% Success	% Success
Risser test	0	56.6	75
	1	60	95
	2	73.3	81
		*p* = 0.505	*p* = 0.16
Risser Test	0	56.6	75
	1 + 2	65.7	89.5
		*p* = 0.527	*p* = 0.248

**Table 4 children-09-00909-t004:** Clinical and EOS roentgenrogram measures on date 1 (median (IQR)). **Bold indicates *p* < 0.05**.

				MT		ML			
				Failure	Success	*p*	Failure	Success	*p*
				46	42		16	46	
CLINIC			age	12 (11.5, 13)	13 (12, 14)	0.18	13.00 (12, 13)	13.00 (12, 14)	0.067
			Gender = Male (%)	3 (8.6)	5 (9.4)	1	1 (10)	11 (21.2)	0.703
			Standing Height	154 (145, 159)	156 (150, 161)	0.234	152 (147, 155)	158 (154, 166)	**0.01**
			Sitting Height	78 (77, 80)	80 (77, 84)	0.081	78.50 (76.2, 80.7)	82.00 (79, 85)	**0.047**
			Unequal Leg Length (%)	3 (8.6)	4 (7.5)	1	2 (20)	12 (23.1)	1
			Thoracic Rib Hump	15 (10, 20)	12.5 (10, 20)	0.09	5 (5, 6.2)	10 (5, 18.7)	0.253
			Lumbar Prominence	5 (5, 10)	5 (5, 6.25)	0.58	5 (5, 7.5)	100 (5, 10)	0.353
X—RAY	FRONTAL	GLOBAL	Frontal C7 Tilt degree	1.39 (0.5, 2)	1.7 (0.8, 2.4)	0.28	3.56 (2.4, 4.6)	1.33 (0.6, 2.5)	**0.002**
			C7 Tilt mm	9.1 (4.3, 14.1)	12.1 (4.2, 18)	0.3	23.4 (18.3, 33.4)	8.37 (3.1, 15.7)	**<0.001**
			Acromio-Femoral Angle	2.7 (1.8, 4.3)	1.61 (0.8, 3.6)	**0.022**	1.85 (1.4, 2.6)	2.44 (1.5, 3.7)	0.353
		REGIONAL	Main Curve Cobb Angle	29 (24, 33)	25 (22, 28)	**0.008**	28 (22, 33)	24 (20, 27)	0.165
			Main Curve Cobb Angle Subgroup (%)			0.12			0.417
			<15	1 (2.9)	0 (0)		0 (0)	1 (1.9)	
			15–29	22 (62.9)	43 (81.1)		6 (60)	40 (76.9)	
			30–44	12 (34.3)	9 (17.0)		4 (40)	11 (21.2)	
			>45	0 (0)	1 (1.9)				
			Main Curve Vertebrae Number	7 (6, 7)	7 (6, 7)	0.25	5.5 (5, 6)	5 (5, 6)	0.781
			Overcurve Cobb Angle	12.6 (3.7, 17.7)	4.1 (1, 14.2)	**0.021**	17.4 (14, 18.7)	3.4 (1.5, 18.7)	**0.034**
			Undercurve Cobb Angle	17 (13, 21.6)	16.2 (11.6, 19.5)	0.54			
			Iliolumbar Angle	5.4 (0.3, 8.3)	2.9 (0, 7.1)	0.18	12.2 (10.7, 16.1)	10.4 (4.3, 13.6)	0.113
		LOCAL	Main Curve Apex Lateral Displacement	10.9 (5, 21.8)	12.4 (7, 20.3)	0.6	33 (19.9, 37.6)	22.1 (14.4, 25.9)	**0.019**
			Main Curve Apex Vertebrae Wedging	4.4 (2.5, 5.6)	3.4 (2.6, 4.6)	0.3	4.2 (2.2, 5)	3.4 (2.1, 4.6)	0.572
	SAGITTAL	GLOBAL	Sagittal C7 Tilt degree	4.4 (3.0, 5.7)	2.1 (0.6, 4.7)	0.001	5.2 (1.1, 7.2)	3.3 (1.8, 5)	0.45
		REGIONAL	Lumbar Lordosis	57.9 (48.9, 62.9)	58.5 (51.6, 64.6)	0,46	62.4 (55.9, 74.5)	58.9 (52.5, 64.7)	0.164
			Lumbar Lordosis Vertebrae Number	5 (4, 5)	5 (3, 5)	0.54	5 (4, 5)	5 (4, 5)	0.927
			Thoracic Kyphosis	31.5 (26.7, 39.5)	31.7 (23.7, 38.3)	0.35	39 (34.9, 42.6)	37 (31.2, 42.7)	0.75
			Thoracic Kyphosis Vertebrae Number	10 (9, 11)	10 (8, 11)	0.27	10 (8.2, 11)	10 (9, 11)	0.95
		PELVIS	Pelvic Incidence	44.8 (35.1, 52.6)	48.7 (40.3, 59.6)	0.13	57 (45.3, 74.8)	49.6 (38.7, 58)	0.32
			Sacral Slope	39.7 (35.2, 44.9)	39.6 (34, 45.2)	0.51	40.7 (34.5, 50.2)	38.4 (33.7, 45.3)	0.54
			Pelvic Tilt	6.1 (1.1, 8.5)	9.5 (3.6, 13.1)	0.045	12 (3.6, 18.6)	8.8 (3, 13)	0.23

**Table 5 children-09-00909-t005:** Inbrace EOS Roentgenogram measures date 2 (median (IQR)). **Bold indicates *p* < 0.05**.

			MT			ML		
			Failure	Success	*p*	Failure	Success	*p*
		n	46	42		16	46	
FRONTAL	GLOBAL	Frontal C7 Tilt degree	1.4 (0.5, 2)	1.7 (0.8, 2.4)	0.28	2.6 (1.7, 3.3)	1.3 (0.8, 3)	0.14
	REGIONAL	Main Curve Cobb Angle	17.3 (12.3, 23.1)	3.5 (9, 16.4)	**0.01**	15.9 (8.1, 17.9)	8.6 (4.9, 14)	**0.031**
		Main Curve Cobb Angle Reduction	35% (23%, 53%)	46% (38%, 62%)	**0.022**	42% (19%, 54%)	66% (44%, 84%)	**0.028**
	LOCAL	Main Curve Apex Lateral Displacement	9.52 (4.2, 12.9)	7.7 (4, 11)	0.32	17 (6.4, 23.3)	8.6 (4, 15)	0.16
SAGITTAL	GLOBAL	Sagittal C7 Tilt	5.1 (3.4)	2.8 (2.7)	**0.001**	3.8 (2.5, 5.2)	3.6 (1.1, 5.7)	0.98
	REGIONAL	Lumbar Lordosis	52.3 (48.6, 69)	55 (48.9, 61.4)	0.42	62.4 (55.9, 74.5)	58.9 (52.5, 64.7)	0.16
		Lumbar Lordosis Vertebrae Number	5 (4, 6)	5 (4, 6)	0.074	5 (4, 5)	5 (4, 5)	0.93
		Thoracic Kyphosis	23.6 (16.1, 28.2)	25.3 (15.6, 29.5)	0.81	39 (34.9, 42.6)	37 (31.2, 42.7)	0.75
		Thoracic Kyphosis Vertebrae Number	10 (9, 11)	10 (8, 11)	0.33	10 (8.2, 11)	10 (9, 11)	0.95
	PELVIS	Pelvic Incidence	43.7 (38.7, 52.5)	50.4 (41.5, 57.9)	0.11	57 (45.3, 74.8)	49.6 (38.7, 58)	0.32
		Sacral Slope	39.6 (33.9, 43.8)	41.7 (35.9, 47.3)	0.30	40 (39, 55.1)	38.3 (33.8, 46.4)	0.27
		Pelvic Tilt	7.2 (3.4, 11.4)	8.9 (4.9, 14.3)	0.15	14.4 (10.2, 19.6)	9.9 (4.3, 12.9)	0.072

**Table 6 children-09-00909-t006:** Main thoracic multivariate analysis. **Bold indicates *p* < 0.05**.

		95% Confidence Interval		
	Odds Ratio	2.50%	97.50%	*p*
Age	0.86	0.41	1.72	0.67
Sitting Height	1.15	0.94	1.44	0.18
Thoracic Rib Hump	0.97	0.86	1.08	0.55
Risser Test	1.41	0.45	5.09	0.57
Acromio-Femoral Angle	0.82	0.50	1.32	0.41
OverCurve CobbAngle	0.92	0.83	0.99	**0.044**
Main Curve Apex Vertebrae Wedging	1.22	0.88	1.73	0.24
Pelvic incidence	0.91	0.76	1.08	0.30
Pelvic Tilt	1.19	0.97	1.53	0.13
Lumbar Lordosis	1.07	0.91	1.26	0.42
Thoracic Kyphosis	1.03	0.94	1.12	0.53
Inbrace Sagittal C7 Tilt (degrees)	0.72	0.54	0.91	**0.014**
Inbrace Lumbar Lordosis Vertebrae Number	0.77	0.49	1.14	0.21
Bad Compliance	0.26	0.02	2.16	0.23
Main Curve CobbAngle Reduction	11.44	0.36	56.74	0.19

**Table 7 children-09-00909-t007:** Main lumbar multivariate analysis. **Bold indicates *p* < 0.05**.

		95% Confidence Interval		
	Odds Ratio	2.5%	97.5%	*p*
Age	0.90	0.41	1.93	0.79
Standing Height	1.42	1.07	2.12	**0.035**
Sitting Height	0.68	0.39	1.09	0.14
Frontal C7 Tilt	0.43	0.19	0.83	**0.02**
In-brace Apex Lateral displacement	0.90	0.79	1.00	0.07

## Data Availability

All data are available.

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
