# Peer review of "Adolescent and Juvenile Idiopathic Scoliosis: Which Patients Obtain Good Results with 12 Hours of Cheneau–Toulouse–Munster Nighttime Bracing?"

_children, 2022, doi:10.3390/children9060909_

Round 1

Reviewer 1 Report

The study entitled “Adolescent and Juvenile Idiopathic Scoliosis: what patient obtains good results with 12 hours Cheneau-Toulouse-Munster nighttime bracing?” aims to analyze the efficiency of 12h nighttime CTM brace-wear on adolescent idiopathic scoliosis. Secondary objectives were to find influencing factors of good results.

Comments and questions:

  1. Please define specific medical terms about scoliosis and bracing in this manuscript, and briefly describe the measurement methods as well, such as frontal C7 tilt, sagittal C7 tilt, or acromio-femoral angle, etc.

  1. In line 12 (OBJECTIVE: to analyze the efficiency of 12h nighttime…), to of “t”o should be uppercased.

  1. In Line 184 and 185, the statement: “Higher overcurve-CA was strongly associated with failure to with OR=0.42 IC95%0.54;0.91 p=0.0437. (table 6)” seems not to have its origin in the table 6. Please declare that.

  1. The tables seem to have too many “numbers” and not enough notes, and hard to read.

  1. Regarding the main thoracic curve, the “conclusion” in the abstract seems not completely consistent with “conclusions” in the manuscript.

  1. The authors mentioned the previous study found poorer success ratios for a wearing time down to 12h (71% of success) and <7h (31%) with Boston brace. Why did the authors choose 12 hours of wearing time? Please clearly describe the reason.

  1. How to monitor the 12h nighttime wearing time? Please describe clearly. (Why emphasizes the nighttime?)

  1. The authors said that 12% of our patients were bad compliers and 55% of them were in failure group (p=0.025). Compliance seems one of the effective factors. However, there was 94% compliance for 8h wearing time. Why do the authors perform 12h instead of 8 h?

Author Response

  • Please define specific medical terms about scoliosis and bracing in this manuscript, and briefly describe the measurement methods as well, such as frontal C7 tilt, sagittal C7 tilt, or acromio-femoral angle, etc.

Reply: We have added descriptions in the text and a supplemental file with figures.

  • In line 12 (OBJECTIVE: to analyze the efficiency of 12h nighttime…), to of “t”o should be uppercased.

Reply: We have corrected as suggested.

  • In Line 184 and 185, the statement: “Higher overcurve-CA was strongly associated with failure to with OR=0.42 IC95%0.54;0.91 p=0.0437. (table 6)” seems not to have its origin in the table 6.  Please declare that.

Reply: We have corrected as suggested.

  • The tables seem to have too many “numbers” and not enough notes, and hard to read.

Reply: Thank you for this comment. We have simplified the tables.

  • Regarding the main thoracic curve, the “conclusion” in the abstract seems not completely consistent with “conclusions” in the manuscript.

Reply: Thank you for this comment. We modified the conclusions to assure consistency.

  • The authors mentioned the previous study found poorer success ratios for a wearing time down to 12h (71% of success) and <7h (31%) with Boston brace. Why did the authors choose 12 hours of wearing time? Please clearly describe the reason.

Reply: 12 h is consistent with the most important study about brace by Weinstein et al. (ref. 7)

  • The authors said that 12% of our patients were bad compliers and 55% of them were in failure group (p=0.025). Compliance seems one of the effective factors. However, there was 94% compliance for 8h wearing time. Why do the authors perform 12h instead of 8 h?

Reply: 12 h is consistent with the most important study about brace by Weinstein et al. (ref. 7).

  • How to monitor the 12h nighttime wearing time? Please describe clearly.

Reply: At each visit, we asked parents and patients to honestly declare at what time the patients use to wear the brace and at what time they take off.

  •  Why emphasizes the nighttime?

Reply: this is related to the fact that night-time growth is higher than day-time. Moreover, from our experience, it is psychologically easier for teens to wear a brace at home than at school.  Thank you for reviewing!

Reviewer 2 Report

Please check spelling and capital letters throughout the manuscript.

I would change Risser Test to Risser sign.

Inclusion criteria Cobb>30 does not match result s with average Cobb angle 26.

The result and discussion parts are difficult to read. 

However, a great study. Thank you!

Author Response

Please check spelling and capital letters throughout the manuscript.

Reply: We have checked spelling and capitals. Moreover, a native English speaker has edited the text.

I would change Risser Test to Risser sign.

Reply: We have corrected as suggested.

Inclusion criteria Cobb>30 does not match result s with average Cobb angle 26.

Reply: indeed, inclusion criteria were angle increase or angle >30°. We have corrected this point.

The result and discussion parts are difficult to read. 

Reply: Thank you for this comment. We have tried to simplify these parts.

However, a great study. Thank you!

Reply: Thank you for reviewing!

Round 2

Reviewer 1 Report

After revision the article because more readable. It can be accepted for publication in the Journal.